

# A Sparse-Modeling Based Approach for Class Specific Feature Selection

Davide Nardone, Angelo Ciaramella and Antonino Staiano

Dipartimento di Scienze e Tecnologie, Università degli Studi di Napoli "Parthenope", Naples, Italy

## ABSTRACT

In this work, we propose a novel Feature Selection framework called Sparse-Modeling Based Approach for Class Specific Feature Selection (SMBA-CSFS), that simultaneously exploits the idea of Sparse Modeling and Class-Specific Feature Selection. Feature selection plays a key role in several fields (e.g., computational biology), making it possible to treat models with fewer variables which, in turn, are easier to explain, by providing valuable insights on the importance of their role, and likely speeding up the experimental validation. Unfortunately, also corroborated by the no free lunch theorems, none of the approaches in literature is the most apt to detect the optimal feature subset for building a final model, thus it still represents a challenge. The proposed feature selection procedure conceives a two-step approach: (a) a sparse modeling-based learning technique is first used to find the best subset of features, for each class of a training set; (b) the discovered feature subsets are then fed to a class-specific feature selection scheme, in order to assess the effectiveness of the selected features in classification tasks. To this end, an ensemble of classifiers is built, where each classifier is trained on its own feature subset discovered in the previous phase, and a proper decision rule is adopted to compute the ensemble responses. In order to evaluate the performance of the proposed method, extensive experiments have been performed on publicly available datasets, in particular belonging to the computational biology field where feature selection is indispensable: the acute lymphoblastic leukemia and acute myeloid leukemia, the human carcinomas, the human lung carcinomas, the diffuse large B-cell lymphoma, and the malignant glioma. SMBA-CSFS is able to identify/retrieve the most representative features that maximize the classification accuracy. With top 20 and 80 features, SMBA-CSFS exhibits a promising performance when compared to its competitors from literature, on all considered datasets, especially those with a higher number of features. Experiments show that the proposed approach may outperform the state-of-the-art methods when the number of features is high. For this reason, the introduced approach proposes itself for selection and classification of data with a large number of features and classes.

# INTRODUCTION

Data analysis is the process of evaluating data, that is often subject to high-dimensional feature spaces, i.e., where data are represented in, whatever the area of study, from biology to pattern recognition to computer vision. High dimensionality often translates into

Corresponding author
Davide Nardone,
davide.nardone@live.it

over-fitting, large computational costs and poor performance thus getting a learning task in trouble. Consequently, high-dimensional feature spaces need to be lowered since its feature vectors are generally uninformative, redundant, correlated to each other and also noisy. In this paper, we focus on feature selection, which is undertaken to identify discriminative features by eliminating the ones with little or no predictive information, based on certain criteria, in order to treat with data in low dimensional spaces.

Feature Selection (FS) is the process of selecting a subset of relevant features to use in model construction. FS plays a key role in computational biology, for instance, microarray data analysis involves a huge number of genes with respect to (w.r.t.) a small number of samples, and effectively identifying the most significant differentially expressed genes under different conditions is prominent (*Xiong, Fang & Zhao, 2001*). The selected genes are very useful in clinical applications such as recognizing diseased profiles (*Calcagno et al., 2010*; *Staiano et al., 2013*; *Di Taranto et al., 2015*; *Camastra, Di Taranto & Staiano, 2015*), nonetheless, because of its high costs, the number of experiments that can be used for classification purposes is usually limited due to the small number of samples compared to the large number of genes in an experiment, that gives rise to the *Curse of Dimensionality* problem (*Friedman, Hastie & Tibshirani, 2001*), which challenges the classification as well as other data analysis tasks (*Staiano et al., 2004*; *Ciaramella et al., 2008*). Furthermore, microarray data are usually not immune from several issues, such as sensitivity, accuracy, specificity, reproducibility of results, and noisy data (*Draghici et al., 2006*). For these reasons, it is unsuitable to use microarray data as they are; however, after several corrections, the relevant genes can be selected by FS approaches, and for instance use Real-Time PCR (*Xiong, Fang & Zhao, 2001*) to validate the results.

Taking a look at the literature, by *googling* the keyword ''*feature selection*'', one gets lost in an ocean of techniques (the reader may refer to classical reviews in *Saeys, Inza & Larrañaga (2007)*, *Guyon & Elisseeff (2003)*, *Hoque, Bhattacharyya & Kalita (2014)* on the topic), often designed to tackle a specific data set. The reasons for the abundance of techniques are in the heterogeneity of the available scientific data sets and also by the limitations dictated by *no free lunch theorems* (*Wolpert & Macready, 1997*), determining the existence of no general-purpose technique which is well suited to a plethora of different kind of data. A typical taxonomy organizes FS techniques (*Jović, Brkić & Bogunović, 2015*) in three main categories, namely *filter*, *wrapper* and *embedded* methods, whose belonging algorithms select a single feature subset from a complete list of features. Another perspective instead, divides FS techniques in two classes, namely, Traditional Feature Selection (TFS) for all classes (that includes filter, wrapper and embedded methods mentioned so far), and Class-Specific Feature Selection (CSFS) (*Fu & Wang, 2002*). Usually, a TFS algorithm selects one subset of features for all classes although it may be not the best one for some classes, thus leading to undesirable results. Differently, a CSFS policy permits to select a distinct subset of features for each class, and it can use any traditional *feature selector*, for choosing, given the set of classes of a classification problem, one distinct grouping of features for each class. Depending on the type of the feature selector, the overall process may slightly change. Nevertheless, it is worth pointing out that a CSFS scheme heavily depends on the use of a specific classifier, while its use should be independent of both the classifier

of the classification step and the feature selector strategy. To this end, a General Framework CSFS has been proposed in (*Pineda-Bautista, Carrasco-Ochoa & Martınez-Trinidad, 2011*) which allows using any traditional feature selector as well as any classifier.

In this paper, on the basis of the general framework for CSFS, we propose a novel strategy to FS, namely a Sparse-Modeling Based Approach for Class-Specific Feature Selection, consisting of a two-step procedure. Firstly, a sparse modeling based learning technique is used to find the best subset of features for each class of the training set. In doing so, it is assumed that a class is represented by using a subset of features, called *representatives*, such that each sample in a specific class, can be described as a linear combination of them. Secondly, the discovered feature subsets are fed to a class-specific feature selection scheme in order to assess the effectiveness of the selected features in classification tasks. To this end, an ensemble of classifiers is built by training a given classifier, one for each class, on its own feature subset, i.e., the one discovered in the previous step, and a proper decision rule is adopted to compute the ensemble responses. In this way, the dilemma of choosing specific TFS strategy and classifiers in the CSFS framework is effectively mitigated.

## METHODS

The sparse-modeling based approach for class-specific feature selection, is based on the concepts of sparse modeling and class-specific feature selection that need to be properly introduced.

### Sparse Modeling fundamentals

An active developing field of statistical learning is focused around the notion of sparsity (*Tibshirani, 1994*; *Ciaramella & Giunta, 2016*). A Sparse Model (SM) is a model that can be much easier to estimate and interpret than a dense model. The sparsity assumption allows extracting meaningful features from large data sets. The aim of the first phase of the proposed approach is to use a sparse modeling for finding data representatives without any transformation and to be performed directly in the data space. In other words, we wish to find a ranking of the most representative features that best reconstruct the data collection. Most approaches are based on a $l_1$-norm regularization such as LASSO (*Tibshirani, 1994* and Sparse Dictionary Learning *Elhamifar, Sapiro & Vidal, 2012*). Formally, given a set of features in $\mathbb{R}^m$ arranged as columns of a data matrix $\mathbf{X} = [\mathbf{x}_1,\ldots,\mathbf{x}_n]$, the task is to find representative features given a fixed feature space belonging to a collections of data points (see *Mairal et al., 2008*; *Aharon, Elad & Bruckstein, 2006*; *Engan, Aase & Husoy, 1999*; *Jolliffe, 1986*; *Ramirez, Sprechmann & Sapiro, 2010*). That task can conveniently be described in the *Dictionary Learning* (DL) framework, where the aim is to simultaneously learn a compact dictionary $\mathbf{D} = [\mathbf{d}_1,\ldots,\mathbf{d}_k] \in \mathbb{R}^{m \times k}$ and coefficients $\mathbf{C} = [\mathbf{c}_1,\ldots,\mathbf{c}_n] \in \mathbb{R}^{k \times n}$, with $k \ll n$, that can well represent collections of data points (*Ciaramella, Gianfico & Giunta, 2016*). The best representation of the data is obtained by minimizing the following objective function

$$\sum_{i=1}^{n} \|\mathbf{x}_i - \mathbf{D}\mathbf{c}_i\|_2^2 = \|\mathbf{X} - \mathbf{D}\mathbf{C}\|_F^2 \tag{1}$$

w.r.t. the dictionary $\mathbf{D}$ and the coefficient matrix $\mathbf{C}$, subject to appropriate constraints.

However, the dictionary learned atoms almost never correspond to the original feature space (*Aharon, Elad & Bruckstein, 2006*; *Ramirez, Sprechmann & Sapiro, 2010*; *Mairal et al., 2009*). In order to find a subset of features that best represent the entire feature space, the optimization problem in Eq. (1) is reformulated forcing the dictionary $\mathbf{D}$ to be the data matrix $\mathbf{X}$ (*Elhamifar, Sapiro & Vidal, 2012*):

$$\sum_{i=1}^{n} \|\mathbf{x}_i - \mathbf{X}\mathbf{c}_i\|_2^2 = \|\mathbf{X} - \mathbf{X}\mathbf{C}\|_F^2, \tag{2}$$

where $F$ is the Frobenius norm. Equation (2) is minimized w.r.t the coefficient matrix $\mathbf{C} \triangleq [\mathbf{c}_1, \ldots, \mathbf{c}_n] \in \mathbb{R}^{n \times n}$, subject to additional constraints. In other words, the *reconstruction error* of each feature component is minimized by linearly combining all the components of the feature space. To choose $k \ll n$ representatives involved in the linear reconstruction of each component in Eq. (2), the following constraint is added to the model

$$\|\mathbf{C}\|_{0,q} \leq k, \tag{3}$$

where the mixed $\ell_0/\ell_q$ norm is defined as $\|\mathbf{C}\|_{0,q} \triangleq \sum_{i=1}^{N} I(\|\mathbf{c}^i\|_q > 0)$, $\mathbf{c}^i$ denotes the $i$-th row of $\mathbf{C}$, and $I(\cdot)$ denotes the indicator function. In a nutshell, $\|\mathbf{C}\|_{0,q}$ counts the number of nonzero rows of $\mathbf{C}$. The indices of the nonzero rows of $\mathbf{C}$ correspond to the indices of the columns of $\mathbf{X}$ which are chosen as the representative features. Since the aim is to select $k \ll n$ representative features that can reconstruct each feature of the $\mathbf{X}$ matrix up to a fixed error, the optimization problem to solve is

$$\underset{\mathbf{C}}{\text{minimize}} \quad \|\mathbf{X} - \mathbf{X}\mathbf{C}\|_F^2$$

$$\text{subject to} \quad \|\mathbf{C}\|_{0,q} \leq k, \mathbf{1}^T \mathbf{C} = \mathbf{1}^T \tag{4}$$

where $\mathbf{1}^T \mathbf{C} = \mathbf{1}^T$ is the affine constraint for selecting representatives that are invariant w.r.t. a global translation of the data (as requested by dimensionality reduction methods). This is an NP-hard problem as it implies a combinatorial calculation over every subset of the $k$ columns of $\mathbf{X}$. Therefore, relaxing $\ell_0$ to $\ell_1$ norm, the problem becomes

$$\underset{\mathbf{C}}{\text{minimize}} \quad \|\mathbf{X} - \mathbf{X}\mathbf{C}\|_F^2$$

$$\text{subject to} \quad \|\mathbf{C}\|_{1,q} \leq \tau, \mathbf{1}^T \mathbf{C} = \mathbf{1}^T \tag{5}$$

where $\|\mathbf{C}\|_{1,q} \triangleq \sum_{i=1}^{N} \|\mathbf{c}^i\|_q$ is the sum of the $\ell_q$ norms of the rows of $\mathbf{C}$ and $\tau > 0$ is an appropriate chosen parameter. The solution of the optimization (Eq. (5)) not only provides the representative features as the nonzero rows of the $\mathbf{C}$, but also provides information about the ranking of the selected features. More precisely, a representative that has higher ranking takes part in the reconstruction process more than the others, hence, its corresponding row in the optimal coefficient matrix $\mathbf{C}$ has many nonzero elements with large values. Conversely, a representative with lower ranking takes part in the reconstruction process less than the others, hence, its corresponding row in $\mathbf{C}$ has a few nonzero elements with smaller values. Thus, the $k$ representative features $x_{i1}, \ldots, x_{ik}$ are ranked as $i_1 \geq i_2 \geq \cdots \geq i_k$, whenever for the corresponding rows of $\mathbf{C}$ one gets

$$\|\mathbf{c}^{i_1}\|_q \geq \|\mathbf{c}^{i_2}\|_q \cdots \geq \|\mathbf{c}^{i_k}\|_q, \tag{6}$$

---

**Procedure** SMBA

**Input:** X, $N \times M$ matrix where $N$ is the number observations and $M$ is the number of features

$\theta = \{\alpha, \delta, \rho, \eta\}$, parameters vector

**Output:** $I$, set of features selected

1   Variables initialization

3   **while** $\epsilon > \delta$ *and* $t > \rho$ **do**

4     $\beta^{t+1} \leftarrow (X^T X + \rho I)^{-1}$

5     $\theta^{t+1} \leftarrow (S_{\lambda/\rho}(\beta^{t+1} + \mu^t/\rho))$

6     $\mu^{t+1} \leftarrow \mu^t + \rho(\beta^{t+1} - \theta^{t+1})$

7     $\epsilon \leftarrow compute\_error(\beta, \theta)$

8   **end**

9   $I \leftarrow find\_representatives(\theta, \eta)$

---

From a practical point of view, the optimization problem (Eq. (5)) can be expressed by using the Lagrange multipliers

$$\underset{C}{\text{minimize}} \quad \frac{1}{2}\|\mathbf{X} - \mathbf{XC}\|_F^2 + \lambda\|\mathbf{C}\|_{1,q} \quad \text{subject to} \quad \mathbf{1}^T\mathbf{C} = \mathbf{1}^T. \tag{7}$$

In practice, the algorithm is implemented using an Alternating Direction Method of Multipliers (ADMM) optimization framework (*Boyd et al., 2011*). In particular, the features of a given data set are obtained considering representatives of small pairwise coherence features as in a sparse dictionary learning method. It is worth observing the resemblance with the Least Absolute Shrinkage and Selection Operator (LASSO) (*Tibshirani, 1994*). The latter consists of an approach to regression analysis that performs both variable selection and regularization in order to enhance the prediction accuracy and interpretation ability of the statistical model it produces. Recall that the objective of LASSO, in its basic form, is to solve

$$\underset{\beta}{\text{minimize}} \quad \frac{1}{N}\|y - \mathbf{X}\beta\|_2^2$$

$$\text{subject to} \quad \|\beta\|_1 \leq t, \tag{8}$$

where $y = [y_1, \ldots, y_N]$ is the $N$-dimensional vector of outcomes, $\mathbf{X}$ the covariate matrix, $t$ is a free parameter that determines the amount of regularization and $\beta$ is the sparse vector to estimate.

From Eq. (8), one can observe that a sparse matrix can be estimated as in Eq. (7) by considering $\mathbf{X}$ itself as outcome and adding the affine constraint. In the following, the LASSO will be used for classification tasks, adopting a sigmoid function, as it will be described in the experimental setup.

---

**Algorithm 1:** Sparse-Modeling Based Approach for Class-Specific Feature Selection

---

**Input :**    $X = \{x_1, \ldots, x_n\}$ data set

           $y$, class labels

           $\theta$, SMBA parameters

           $m$, maximum number of features to select

           $C$, classifier model (e.g., SVM, KNN, etc)

           $K$, number of folds for performing K-Cross Validation

**Output:**    $\overline{ACM}$, Average Classification Metrics on K folds

1   **begin**
2     $X \leftarrow$ Data standardization
3     $X \leftarrow$ Class balancing(X) by using SMOTE Chawla et al., 2002
4     $X \leftarrow$ Random shuffling(X)
5     Divide X into $K$ folds
6     **foreach** $k_i \in K$ *folds* **do**
7       Set the $k_i$ fold as the test set $X_{test}$
8       Use the remaining *K-1* folds as the train set $X_{train}$
9       Perform the *Class-sample separation* on the train set $X_{train}$
10      *(Note that I is the subset of features selected for each class $c_i \in X_{train}$)*
11      **foreach** $X_{c_i} \in X_{train}$ **do**
12        $I = \{I_{c_i} \ldots I_{c_c}\} \leftarrow \mathrm{SMBA}(X_{c_i}, \theta)$
13      **end**
14      **for** $j \leftarrow 1$ **to** $m$ **do**
15        Build an ensemble classifier $E_j = \{e_{1,j}, \ldots, e_{c,j}\}$ using the *j-th* selected feature $\in I_{c_i}$ and the classifier $C$
16        **foreach** $O \in X_{test}$ **do**
17          $(ACM_j) \leftarrow$ Use $E_j$ to classify the instance $O$
18        **end**
19        $(ACM) \leftarrow (ACM_j)$
20      **end**
21     **end**
22     $(\overline{ACM}) \leftarrow Average(ACM)$
23   **end**

---

## A Sparse-Modeling Based Approach for Class Specific Feature Selection

A General Framework for Class-Specific Feature Selection (GF-CSFS) is described in (*Pineda-Bautista, Carrasco-Ochoa & Martınez-Trinidad, 2011*). The proposed Sparse-Modeling Based Approach for Class-Specific Feature Selection (SMBA-CSFS) tries to

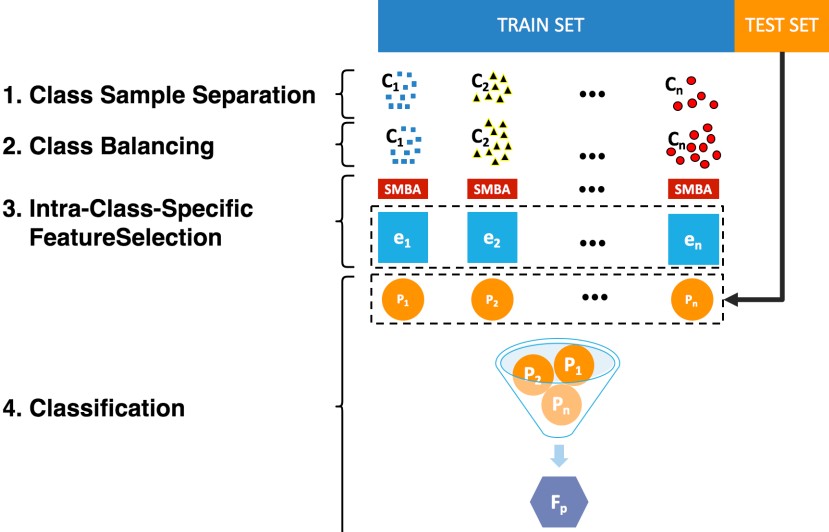

**Figure 1** A Sparse-Modeling Based Approach for Class-Specific Feature Selection.

best represent each class-sample set of an input data set by only using few representative features. More specifically, the method is made up of the following steps:

1. **Class-sample separation:** Unlike the GF-CSFS, SMBA-CSFS does not employ the *Class binarization* stage to transform a $c$-class problem into $c$ binary problems, instead it just uses a simple *Class-sample separation*. Basically, it consists of differentiating the samples among all the classes of the training set for a given data set into several disjoint sets/configurations of samples, one for each class (See Fig. 1).

2. **Class balancing:** Once the class sample set of the training set has been split apart (by applying the above *Class-sample separation* step), it may be possible that each class-subset results unbalanced. Therefore, the SMOTE (*Chawla et al., 2002*) re-sampling method is applied to balance each class-subset. Technically speaking, it is important to point out that steps 1–2 are interchangeable, meaning that there are no differences in doing the first one before the other.

3. **Intra-Class-Specific feature selection:** The *Sparse-Modeling Based Approach* is used for retrieving, minimizing Eq. (7), the most representative features for each class-sample set of the training set that best represent/reconstruct the whole class of objects. In doing so, the approach takes advantage of the intra-class properties for selecting the best feature subset (describing each class) which is used to improve the classification accuracy against TFS and GF-CSFS.

4. **Classification:** Since the training set gets split into different class-sample subsets, we embraced the idea of using a wise-ensemble procedure for training a classification model for discriminating new incoming instances. As in *Pineda-Bautista, Carrasco-Ochoa & Martınez-Trinidad (2011)*, given a class $c_i$, a classifier $e_i$ is trained on the original data set only using the selected features for $c_i$, for $i = 1, \ldots, c$. Overall, an ensemble classifier $E = \{e_1, \ldots, e_c\}$ is constructed. In order to classify a new instance $O$ through

the ensemble, the natural dimension of $O$ needs to be lowered to the dimension $d_i$ of the classifier $e_i, i = 1 \ldots, c$. This way, for determining to which class $O$ belongs to, an *ad-hoc majority rule* is used:

(a) If a classifier outputs the same class for which the features, used for training $e_i$ were selected, i.e., the $e_i$ output is $c_i$, then $O$ belongs to $c_i$. In case of a tie, i.e., when several classifiers respond $c_i$, a *majority vote* is needed among all classifiers to determine the class of $O$. If still a tie occurs, $O$ will belong to the class that received more votes among the tied classes.

(b) If no classifier outputs the class whose selected features are used for training $e_i$ belongs to the class winning the majority voting. If there is a tie, then $O$ will belong to the class that received more votes among the tied classes.

Finally, since a recursive tie may occur, in that case, the instance $O$ would be classified as $c_i$ by randomly choosing a class among all the tied classes. The algorithm in Fig. 1, illustrates the pseudo-code describing the CSFS-SMBA procedure. Basically, it first standardizes, *class-balances* and shuffles the data set $X$, then divides it into $k$ folds, assigning the $k_i$-th fold as test set $X_{test}$ and the remaining $K - 1$ folds as train set $X_{train}$. The algorithm iteratively performs the task of *class-sample separation*, to split the sample belonging to different classes $X_{c_i}$, on which the algorithm 1 (illustrated in page 4) is performed to output the $m$ most representative features for each class (line 12). The selected features are first used, one at time, for training an ensemble classifier $E_j$, and later for classifying each instance $O$ belonging to the test set $X_{test}$. Finally, for all the ensemble models up to $m$ selected features, the algorithm outputs the $\overline{ACM}$ matrix, storing several model evaluation metrics.

## EXPERIMENTAL RESULTS

In the experiments, the SMBA-CSFS performance have been assessed on nine publicly available microarray data sets. The classifiers used to determine the goodness of the selected feature subsets are a Support Vector Machine (SVM) with a linear kernel and parameter $C = 1$, a Naive Bayes, a K-Nearest Neighbors (KNN) using $k = 5$, and a Decision Tree.

### Data sets description

In order to validate the introduced approach, a number of data sets exemplifying the typical data processing in the biological field are used in the experiments. In the following, a brief description of all the data sets employed in the experiments.

1. The **ALLAML** data set (*Golub et al., 1999*) contains in total 72 samples in 2 classes: ALL and AML, which have 47 and 25 samples, respectively. Every sample contains 7,129 gene expression values.

2. The **LEUKEMIA** data set (*Golub et al., 1999*) contains in total 72 samples in 2 classes: acute lymphoblastic and acute myeloid. It is a modified version of the original ALLAML data set, where the original baseline genes (7,129) were cut off before further analysis. The number of genes that are used in the binary classification task is 7,070.

3. The **CLL_SUB_111** data set (*Haslinger et al., 2004*) has gene expressions from high density oligonucleotide arrays containing genetically and clinically distinct subgroups

of B-cell chronic lymphocytic leukemia (B-CLL). The data set consists of $11,340$ attributes, 111 instances and 3 classes.

4. The **GLIOMA** data set (*Nutt et al., 2003*) contains in total 50 samples in 4 classes: cancer glioblastomas, non-cancer glioblastomas, cancer oligodendrogliomas and non-cancer oligodendrogliomas, which have $14, 14, 7, 15$ samples, respectively. Each sample has $12,625$ genes. After a preprocessing, the data set has been shrunk to 50 samples and $4,433$ genes.

5. The **LUNG** data set (*Bhattacharjee et al., 2001*) contains in total 203 samples in 5 classes: adenocarcinomas, squamous cell lung carcinomas, pulmonary carcinoids, small-cell lung carcinomas and normal lung, with $139, 21, 20, 6, 17$ samples, respectively. The genes with standard deviations smaller than 50 expression units were removed getting a data set with 203 samples and $3,312$ genes.

6. The **LUNG_DISCRETE** data set (*Peng, Long & Ding, 2005*) contains 73 samples in 7 classes where, each sample consists of 325 gene expressions. The cardinalities of each sample in the LUNG_DISCRETE data set are $6, 5, 5, 16, 7, 13, 21$, respectively.

7. The **DLBCL** data set (*Alizadeh et al., 2000*) is a modified version of the original DLBCL data set. It consists of 96 samples in 9 classes, where each sample is defined by the expression of $4,026$ genes. The cardinalities of each sample in the DLBCL data set are $46, 10, 9, 11, 6, 6, 4, 2, 2$, respectively.

8. The **CARCINOM** data set (*Su et al., 2001*) contains 174 samples in 11 classes: prostate, bladder/ureter, breast, colorectal, gastroesophagus, kidney, liver, ovary, pancreas, lung adenocarcinomas and lung squamous cell carcinoma, with $26, 8, 26, 23, 12, 11, 7, 27, 6, 14, 14$ samples, respectively. After a preprocessing as described in *Yang et al. (2006)*, the data set has been shrunk to 174 samples and $9,182$ genes.

9. The **GCM** data set (*Ramaswamy et al., 2001*) contains 190 samples in 14 classes: breast, prostate, lung, colorectal, lymphoma, bladder, melanoma, uterus, leukemia, renal, pancreas, ovary, mesothelioma and central nervous system, where each sample consist of 16,063 gene expression signatures. The cardinalities of each sample in the data set are $11, 11, 20, 11, 30, 11, 22, 10, 11, 11, 11, 10, 11, 10$, respectively.

All data sets are available at the following data repository (*Nardone, Ciaramella & Staiano, 2019a*). All the information about the data sets are summarized in Table 1.

## Experiment setup

To validate the effectiveness of the SMBA-CSFS model, it has been compared against several TFS and the GF-CSFS proposed in *Pineda-Bautista, Carrasco-Ochoa & Martínez-Trinidad (2011)*. SMBA-CSFS is firstly compared against TFS methods and, since the framework in *Pineda-Bautista, Carrasco-Ochoa & Martínez-Trinidad (2011)* can use any TFS method as base for performing CSFS, some experiments using both filter and wrapper methods (injection process) were made. In addition, the accuracy results were also compared against those obtained on the basis of all the features (BSL). The following TFS methods have been chosen for comparing purposes:

**Table 1  Data sets description.**

|  | Size | # Features | # Classes |
|---|---|---|---|
| ALLAML | 72 | 7,129 | 2 |
| LEUKEMIA | 72 | 7,070 | 2 |
| CLL_SUB_111 | 111 | 11,340 | 3 |
| GLIOMA | 50 | 4,434 | 4 |
| LUNG_C | 203 | 3,312 | 5 |
| LUNG_D | 73 | 325 | 7 |
| DLBCL | 96 | 4,026 | 9 |
| CARCINOM | 174 | 9,182 | 11 |
| GCM | 190 | 16,063 | 14 |

- **LASSO** (*Tibshirani, 1994*): LASSO method involves penalizing the absolute size of the regression coefficients and it is usually used for creating parsimonious models in presence of a *large* number of features. The model implemented is a modified version of the classical LASSO, adapted for classification purposes. In particular, in Eq. (8), the product $\mathbf{X}\beta$ is transformed by a sigmoid function in order to address the classification problem.
- **EN** (*Zou & Hastie, 2005*): Elastic Net is a hybrid of ridge regression and LASSO regularization. Like LASSO, Elastic Net can generate reduced models by achieving zero-valued coefficients. Experimental studies have suggested that the Elastic Net technique can outperform LASSO on data with highly correlated features. As for LASSO, a modified version adapted for classification purposes has been implemented.
- **RFS** (*Nie et al., 2010*): Robust Feature Selection method is a sparse based-learning approach for feature selection which emphasizes the joint $\ell_{2,1}$ norm minimization on both loss and regularization function.
- **ls-$\ell_{2,1}$** (*Tang, Alelyani & Liu, 2014*): ls-$\ell_{2,1}$ is a supervised sparse feature selection method. It exploits the $\ell_{2,1}$-norm regularized regression model for joint feature selection, from multiple tasks where the *classification objective function* is a quadratic loss.
- **ll-$\ell_{2,1}$** (*Tang, Alelyani & Liu, 2014*): ll-$\ell_{2,1}$ is a supervised sparse feature selection method which uses the same concept of ls-$\ell_{2,1}$ but instead uses a *logistic loss* as *classification objective function*.
- **Fisher** (*Gu, Li & Han, 2012*): Fisher is one of the most widely used supervised filter feature selection methods. It selects each feature as the ratio of inter-class separation and intraclass variance, where features are evaluated independently and, the final feature selection occurs by aggregating the $m$ top ranked ones.
- **Relief-F** (*Kira & Rendell, 1992*; *Kononenko, 1994*): Relief-F is an iterative, randomized and supervised filter approach that estimates the quality of the features according to how well their values differentiate data samples that are near to each other; it does not discriminate among redundant features and performance decreases with few data.
- **mRMR** (*Peng, Long & Ding, 2005*): Minimum-Redundancy-Maximum-Relevance is a mutual information filter based algorithm which selects features according to the maximal statistical dependency criterion.

- **MI** (*Kraskov, Stögbauer & Grassberger, 2004*; *Ross, 2014*): Mutual Information is a non-negative value, which measures the dependency between the variables. Features are selected in a univariate way. The function relies on nonparametric methods based on entropy estimation from k-nearest neighbors distances.
- **SMBA**: Sparse-Modeling Based Approach is nothing else that our SMBA-CSFS model but that only takes into account the SDL strategy for selecting a subset of features considering all the classes in the feature selection process.

We pre-processed all the data sets by using the *Z-score* (*Kreyszig, 2010*) normalization. To fairly compare the considered supervised feature selection methods, we have firstly tuned the parameters for all methods by using a "grid-search" strategy (*Tang, Alelyani & Liu, 2014*) and finally, for evaluating the performance of all the methods, it has been considered a number of features ranging from 1 to 80 by performing a 5-fold Cross Validation (CV).

The performance of the classification algorithms among all the methods have been evaluated by using the metrics of Accuracy along with the standard deviations (ACC ± STD), Precision (P), Recall (R) and F-measure (F), which are computed as illustrated in *Sokolova & Lapalme (2009)*. In addition, to give a better and summarized understanding between the performance of the models, we also computed the Area Under the Curve (AUC) and the Receiver Operating Characteristic (ROC) curves, where the former is a useful tool for evaluating the quality of class separation for a classifier while the latter makes it easier to compare the ROC curve of one model to another.

## DISCUSSION

The experiments have been performed on a workstation with a dual Intel(R) Xeon(R) 2.40 GHz and 64GB RAM. The developed code is available at *Nardone, Ciaramella & Staiano (2019b)*. For the sake of readability, all the results presented here account only for the SVM classifier, since the performance proved that the proposed approach is a little sensitive to the choice of a specific classifier (indeed, the performance of each classifier are rather comparable). Nevertheless, the interested reader may refer to the Supplemental Material for details on additional results concerning all the used classifiers. The experimental results on 5-fold CV for the SVM classifier are summarized in Tables 2–5. Figures 2–5 show all the accounted model evaluation metrics for the ten feature selection methods on the nine considered data sets.

We compared the performance of our method against TFS methods (see Tables 2–3) and GF-CSFS framework (see Tables 4–5). By looking at accuracy, precision, recall and F-measure, SMBA-CSFS is able to better discriminate among the classes of the LUNG_C, LUNG_D, CARCINOM, DLBLC and GCM data sets in most of the cases, when top 20 and 80 features are considered. In this latter case, when SMBA-CSFS performs worse then its competitors, the corresponding performance tend to be comparable. On the remaining data sets, each with a number of classes less than 5, namely, ALLAML, LEUKEMIA, CLL_SUB_111 and GLIOMA, SMBA-CSFS is instead outperformed by some of the competitors. Consequently, we can assert that SMBA-CSFS behaves better when working with data sets with many classes (at least 5). One possible reason is due to the

Nardone et al. (2019), *PeerJ Comput. Sci.*, DOI 10.7717/peerj-cs.237

**Table 2  SVM accuracy results (ACC ± STD) on top 20 features using 5-fold CV on different data sets.** TFS methods are compared against our methods (SMBA and SMBA-CSFS). FS: Fisher Score, mRMR: Minimum-Redundancy-Maximum-Relevance, MI: Mutual Information, RFS: Robust Feature Selector, EN: Elastic Net, BSL: all features. The best results are highlighted in bold. The number in parentheses is the number of features when the performance is achieved.

| | Average Accuracy of top 20 features (%) | | | | | | | | |
|---|---|---|---|---|---|---|---|---|---|
| | **ALLAML** | **LEUKEMIA** | **CLL_SUB_111** | **GLIOMA** | **LUNG_C** | **LUNG_D** | **DLBCL** | **CARCINOM** | **GCM** |
| Fisher | 96.84 ± 0.04(19) | 98.95 ± 0.02(16) | 75.20 ± 0.1(19) | 80 ± 0.04(13) | 91.94 ± 0.02(19) | 91.24 ± 0.1(20) | 97.11 ± 0.02(19) | 65.33 ± 0.05(20) | 94.9 ± 0.00(20) |
| Relief | 95.78 ± 0.04(8) | 97.89 ± 0.03(12) | 76.45 ± 0.03(15) | 80 ± 0.07(19) | 97.12 ± 0.01(20) | 95.2 ± 0.03(14) | 99.76 ± 0.00(20) | 86.52 ± 0.03(18) | 97.14 ± 0.01(20) |
| mRMR | 66.14 ± 0.13(12) | **98.95 ± 0.02(9)** | 71.27 ± 0.1(20) | 66.67 ± 0.1(17) | 95.68 ± 0.013(19) | 95.22 ± 0.02(20) | 99.03 ± 0.01(16) | 89.57 ± 0.04(20) | 97.79 ± 0.01(20) |
| MI | 96.84 ± 0.042(15) | 98.95 ± 0.02(10) | **81.03 ± 0.06(17)** | 78.33 ± 0.04(12) | 97.41 ± 0.014(17) | 94.53 ± 0.03(18) | 98.79 ± 0.01(19) | 93.25 ± 0.05(20) | 95.58 ± 0.01(20) |
| ls-21 | 71.34 ± 0.14(19) | 59.42 ± 0.2(12) | 60.30 ± 0.14(19) | 55 ± 0.07(20) | 92.66 ± 0.05(19) | 93.86 ± 0.04(20) | 92.52 ± 0.01(20) | 66.99 ± 0.03(20) | 96.56 ± 0.01(20) |
| ll-21 | 83 ± 0.11(15) | 88.36 ± 0.06(20) | 73.12 ± 0.06(15) | 0.75 ± 0.12(17) | 98.27 ± 0.015(16) | 93.24 ± 0.04(16) | 94.44 ± 0.02(19) | 83.49 ± 0.03(20) | 97.69 ± 0.01(20) |
| RFS | 87 ± 0.01(15) | 74.33 ± 0.1(18) | 64.73 ± 0.09(15) | 66.67 ± 0.07(17) | 94.10 ± 0.022(20) | 89.77 ± 0.02(19) | 91.06 ± 0.03(18) | 81.85 ± 0.07(18) | 96.77 ± 0.01(20) |
| LASSO | **98.95 ± 0.02(17)** | 71.3 ± 0.08(21) | 68.02 ± 0.06(20) | **83.33 ± 0.05(17)** | 97.99 ± 0.012(16) | 92.51 ± 0.03(12) | **99.52 ± 0.01(16)** | 82.14 ± 0.05(18) | 97.07 ± 0.01(20) |
| EN | **98.95 ± 0.02(17)** | 71.3 ± 0.08(21) | 68.02 ± 0.06(20) | **83.33 ± 0.05(17)** | 97.99 ± 0.012(16) | 92.51 ± 0.03(12) | **99.52 ± 0.01(16)** | 82.14 ± 0.05(18) | 97.07 ± 0.01(20) |
| SMBA | 93.68 ± 0.084(16) | 88.36 ± 0.06(20) | 70.60 ± 0.10(19) | 71.67 ± 0.134(17) | 97.84 ± 0.00(20) | 92.55 ± 0.03(20) | 99.28 ± 0.01(20) | 83.49 ± 0.03(20) | 97.69 ± 0.01(20) |
| SMBA-CSFS | 88.24 ± 0.04(20) | 81.93 ± 0.02(20) | 75.53 ± 0.06(20) | 73.34 ± 0.18(16) | **98.41 ± 0.014(19)** | **97.93 ± 0.03(19)** | 98.30 ± 0.02(13) | **94.95 ± 0.02(19)** | **99.2 ± 0.01(20)** |
| BSL | 97.89 ± 0.04 | 98.95 ± 0.021 | 84.26 ± 0.06 | 85 ± 0.1 | 99.57 ± 0.00 | 98.62 ± 0.02 | 100 ± 0.00 | 98.65 ± 0.01 | 100 ± 0.00 |

Nardone et al. (2019), *PeerJ Comput. Sci.*, DOI 10.7717/peerj-cs.237

**Table 3** **SVM Precision(P), Recall(R) and F-measure(F) on top 20 features using 5-fold CV on different data sets.** TFS methods are compared against our methods (SMBA and SMBA-CSFS). FS: Fisher Score, mRMR: Minimum-Redundancy-Maximum-Relevance, MI: Mutual Information, RFS: Robust Feature Selector, EN: Elastic Net, BSL: all features. The best results are highlighted in bold. The number in parentheses is the number of features when the performance is achieved.

| | ALLAML | | | LEUKEMIA | | | CLL_SUB_111 | | | GLIOMA | | | LUNG_C | | | LUNG_D | | | DLBCL | | | CARCINOM | | | GCM(14) | | |
|---|---|---|---|---|---|---|---|---|---|---|---|---|---|---|---|---|---|---|---|---|---|---|---|---|---|---|---|
| | P | R | F | P | R | F | P | R | F | P | R | F | P | R | F | P | R | F | P | R | F | P | R | F | P | R | F |
| Fisher | 0.98(18) | 0.98(18) | 0.98 | 0.99(15) | 0.99(15) | 0.99 | 0.75(11) | 0.75(11) | 0.75 | 0.68(20) | 0.67(14) | 0.67 | 0.92(19) | 0.92(19) | 0.92 | 0.89(20) | 0.88(15) | 0.88 | 0.9(17) | 0.99(20) | 0.93 | 0.9(19) | 0.89(20) | 0.89 | 0.64(20) | 0.64(20) | 0.64 |
| Relief | 0.96(12) | 0.96(12) | 0.96 | **0.99(4)** | **0.99(4)** | **0.99** | 0.75(17) | 0.75(17) | 0.75 | 0.77(19) | 0.77(19) | 0.77 | 0.97(20) | 0.97(20) | 0.97 | 0.95(20) | 0.95(15) | 0.95 | 0.89(18) | 1.0(20) | 0.94 | 0.89(18) | 0.88(18) | 0.88 | 0.8(20) | 0.8(20) | 0.8 |
| mRMR | 0.8(19) | 0.8(19) | 0.8 | 0.98(6) | 0.98(17) | 0.98 | 0.64(14) | 0.66(14) | 0.65 | 0.7(12) | 0.7(12) | 0.7 | 0.97(20) | 0.97(20) | 0.97 | 0.96(19) | 0.95(19) | 0.95 | 0.95(20) | 0.99(14) | 0.92 | 0.88(20) | 0.91(20) | 0.89 | 0.85(20) | 0.85(20) | 0.85 |
| MI | **0.98(12)** | **0.98(12)** | **0.98** | 0.98(2) | 0.98(2) | 0.98 | **0.76(16)** | **0.76(16)** | **0.76** | 0.74(20) | 0.73(17) | 0.73 | 0.97(20) | 0.97(20) | 0.97 | 0.95(20) | 0.95(20) | 0.95 | 0.95(17) | 0.99(19) | 0.9 | 0.95(17) | 0.95(17) | 0.83 | 0.69(20) | 0.69(20) | 0.69 |
| ls_l21 | 0.83(18) | 0.81(18) | 0.82 | 0.84(20) | 0.82(20) | 0.83 | 0.7(20) | 0.7(20) | 0.7 | 0.7(16) | 0.7(17) | 0.7 | 0.97(20) | 0.97(20) | 0.97 | 0.89(19) | 0.88(19) | 0.88 | 0.81(19) | 0.93(17) | 0.87 | 0.81(19) | 0.81(20) | 0.81 | 0.76(20) | 0.76(20) | 0.76 |
| ll_l21 | 0.92(15) | 0.91(15) | 0.91 | 0.83(20) | 0.83(20) | 0.83 | 0.69(20) | 0.69(20) | 0.69 | 0.65(9) | 0.65(9) | 0.65 | 0.98(18) | 0.98(18) | 0.98 | 0.94(20) | 0.93(19) | 0.93 | 0.92(18) | 0.96(19) | 0.92 | 0.9(17) | 0.86(20) | 0.88 | 0.84(20) | 0.84(20) | 0.84 |
| RFS | 0.86(18) | 0.84(19) | 0.85 | 0.84(20) | 0.76(20) | 0.8 | 0.63(12) | 0.64(12) | 0.63 | 0.71(12) | 0.7(12) | 0.7 | 0.96(19) | 0.96(19) | 0.96 | 0.88(18) | 0.86(18) | 0.87 | 0.89(19) | 0.93(16) | 0.84 | 0.89(18) | 0.84(19) | 0.86 | 0.77(20) | 0.77(20) | 0.77 |
| LASSO | 0.84(20) | 0.84(13) | 0.84 | 0.77(20) | 0.77(20) | 0.77 | 0.71(6) | 0.71(10) | 0.71 | 0.79(14) | **0.78(14)** | **0.78** | 0.94(20) | 0.94(19) | 0.94 | 0.93(19) | 0.9(20) | 0.91 | 0.84(18) | 0.97(19) | 0.9 | 0.84(18) | 0.84(18) | 0.84 | 0.8(20) | 0.8(20) | 0.8 |
| EN | 0.84(20) | 0.84(13) | 0.84 | 0.77(20) | 0.77(20) | 0.77 | 0.71(6) | 0.71(10) | 0.71 | 0.79(14) | **0.78(14)** | **0.78** | 0.94(20) | 0.94(19) | 0.94 | 0.91(19) | 0.9(20) | 0.9 | 0.84(18) | 0.97(19) | 0.9 | 0.84(18) | 0.84(18) | 0.84 | 0.8(20) | 0.8(20) | 0.8 |
| SMBA | 0.9(13) | 0.89(16) | 0.89 | 0.83(20) | 0.83(20) | 0.83 | 0.7(11) | 0.7(11) | 0.7 | 0.68(15) | 0.68(15) | 0.68 | 0.97(18) | 0.97(18) | 0.97 | 0.91(19) | 0.9(19) | 0.9 | 0.92(19) | 0.99(17) | 0.92 | 0.9(19) | 0.86(20) | 0.88 | 0.84(20) | 0.84(20) | 0.84 |
| SMBA-CSFS | 0.83(16) | 0.83(16) | 0.83 | 0.86(20) | 0.86(20) | 0.86 | 0.67(20) | 0.68(20) | 0.67 | 0.8(20) | 0.77(20) | 0.78 | **0.98(15)** | **0.98(15)** | **0.98** | **0.99(19)** | **0.99(19)** | **0.99** | 1.0(20) | 1.0(20) | 1.0 | **0.99(20)** | **0.98(20)** | **0.98** | **0.97(20)** | **0.97(20)** | **0.97** |
| BSL | 1 | 1 | 1 | 1 | 1 | 1 | 0.74 | 0.74 | 0.74 | 0.92 | 0.92 | 0.92 | 0.93 | 0.93 | 0.93 | 0.8 | 0.8 | 0.8 | 1 | 1 | 1 | 0.98 | 0.98 | 0.98 | 1 | 1 | 1 |

Nardone et al. (2019), *PeerJ Comput. Sci.*, DOI 10.7717/peerj-cs.237

**Table 4** **SVM accuracy results (ACC ± STD) on top 20 features using 5-fold CV on different data sets.** GF-CSFS (*Pineda-Bautista, Carrasco-Ochoa & Martınez-Trinidad, 2011*) framework is compared against our SMBA-CSFS. FS: Fisher Score, mRMR: Minimum-Redundancy-Maximum-Relevance, MI: Mutual Information, RFS: Robust Feature Selector, EN: Elastic Net, BSL: all features. The best results are highlighted in bold. The number in parentheses is the number of features when the performance is achieved.

| | Average Accuracy of top 20 features (%) | | | | | | | | |
|---|---|---|---|---|---|---|---|---|---|
| | **ALLAML** | **LEUKEMIA** | **CLL_SUB_111** | **GLIOMA** | **LUNG_C** | **LUNG_D** | **DLBCL** | **CARCINOM** | **GCM** |
| Fisher | **95.90 ± 0.03(13)** | **98.57 ± 0.03(18)** | 80.41 ± 0.02(7) | **82 ± 0.16(17)** | 95.09 ± 0.03(20) | 86.38 ± 0.14(16) | **100 ± 0.00(14)** | 90.86 ± 0.08(20) | 98.98 ± 0.0(18) |
| Relief | 92.95 ± 0.04(5) | 95.81 ± 0.03(10) | **82.41 ± 0.05(12)** | 80 ± 0.19(12) | 91.63 ± 0.02(20) | 86.39 ± 0.07(20) | **100 ± 0.00(11)** | 89.68 ± 0.03(17) | 98.71 ± 0.0(20) |
| mRMR | 75.14 ± 0.09(16) | **98.57 ± 0.03(11)** | 70.69 ± 0.07(12) | 62 ± 0.12(14) | 89.16 ± 0.03(20) | 86.48 ± 0.09(17) | 99.52 ± 0.01(15) | 81.61 ± 0.07(20) | 98.71 ± 0.0(20) |
| MI | 94.38 ± 0.03(18) | 97.14 ± 0.03(4) | 81.03 ± 0.05(20) | **82 ± 0.21(19)** | 95.07 ± 0.015(11) | 79.90 ± 0.18(14) | **100 ± 0.00(19)** | 90.86 ± 0.06(11) | 98.67 ± 0.0(19) |
| ls-21 | 76.47 ± 0.13(6) | 65.52 ± 0.08(3) | 63.44 ± 0.03(20) | 46 ± 0.21(7) | 73.88 ± 0.04(19) | 75.43 ± 0.07(18) | 93.46 ± 0.03(20) | 39.68 ± 0.04(19) | 97.59 ± 0.0(19) |
| ll-21 | 82.1 ± 0.05(16) | 80.67 ± 0.09(15) | 74.58 ± 0.07(20) | 68 ± 0.13(18) | 91.15 ± 0.02(15) | 67.24 ± 0.12(15) | 96.38 ± 0.02(17) | 72.40 ± 0.05(17) | 96.87 ± 0.0(20) |
| RFS | 79.24 ± 0.168(17) | 74.95 ± 0.09(6) | 71.94 ± 0.10(19) | 68 ± 0.21(13) | 82.79 ± 0.05(17) | 68.67 ± 0.07(18) | 96.62 ± 0.01(20) | 58.03 ± 0.18(20) | 96.97 ± 0.01(20) |
| LASSO | 95.73 ± 0.02(6) | 70.3 ± 0.08(15) | 71.29 ± 0.05(18) | 81.67 ± 0.08(19) | 96.26 ± 0.00(18) | 93.22 ± 0.021(20) | **100 ± 0.00(10)** | 87.88 ± 0.03(18) | 96.09 ± 0.0(20) |
| EN | 95.73 ± 0.04(10) | 70.3 ± 0.08(15) | 68.73 ± 0.10(19) | 81.67 ± 0.08(19) | 95.97 ± 0.012(18) | 93.22 ± 0.021(20) | **100 ± 0.00(10)** | 88.56 ± 0.03(19) | 96.09 ± 0.0(20) |
| SMBA-CSFS | 88.24 ± 0.04(20) | 81.93 ± 0.02(20) | 75.53 ± 0.06(20) | 73.34 ± 0.18(16) | **98.41 ± 0.014(19)}** | **97.93 ± 0.03(19)** | 98.30 ± 0.02(13) | **94.95 ± 0.02(19)** | **99.2 ± 0.01(20)** |
| BSL | 97.89 ± 0.04 | 98.95 ± 0.021 | 84.26 ± 0.06 | 85 ± 0.1 | 99.57 ± 0.00 | 98.62 ± 0.02 | 100 ± 0.00 | 98.65 ± 0.01 | 100 ± 0.00 |

Nardone et al. (2019), *PeerJ Comput. Sci.*, DOI 10.7717/peerj-cs.237

**Table 5  SVM Precision(P), Recall(R) and F-measure(F) on top 20 features using 5-fold CV on different data sets.** GF-CSFS (*Pineda-Bautista, Carrasco-Ochoa & Martínez-Trinidad, 2011*) framework is compared against our SMBA-CSFS. FS: Fisher Score, mRMR: Minimum-Redundancy-Maximum-Relevance, MI: Mutual Information, RFS: Robust Feature Selector, EN: Elastic Net, BSL: all features. The best results are highlighted in bold. The number in parentheses is the number of features when the performance is achieved.

| | ALLAML | | | LEUKEMIA | | | CLL_SUB_111 | | | GLIOMA | | | LUNG_C | | | LUNG_D | | | DLBCL | | | CARCINOM | | | GCM(14) | | |
|---|---|---|---|---|---|---|---|---|---|---|---|---|---|---|---|---|---|---|---|---|---|---|---|---|---|---|---|
| | P | R | F | P | R | F | P | R | F | P | R | F | P | R | F | P | R | F | P | R | F | P | R | F | P | R | F |
| Fisher | 0.96(15) | 0.96(14) | 0.96 | 0.97(2) | 0.97(2) | 0.97 | **0.84(4)** | **0.84(4)** | **0.84** | 0.76(8) | 0.75(8) | 0.75 | 0.96(18) | 0.96(18) | 0.96 | 0.97(16) | 0.97(16) | 0.97 | 1.0(17) | 1.0(17) | 1.0 | 0.95(13) | 0.95(13) | 0.95 | 0.93(18) | 0.93(18) | 0.93 |
| Relief | 0.98(16) | 0.98(16) | 0.98 | 0.97(8) | 0.97(8) | 0.97 | 0.82(5) | 0.82(5) | 0.82 | 0.72(19) | 0.7(15) | 0.71 | 0.95(19) | 0.95(19) | 0.95 | 0.96(9) | 0.95(9) | 0.95 | **1.0(10)** | **1.0(10)** | **1.0** | 0.96(17) | 0.96(17) | 0.96 | 0.91(20) | 0.91(20) | 0.91 |
| mRMR | 0.69(8) | 0.69(8) | 0.69 | 0.97(13) | 0.97(4) | 0.97 | 0.84(15) | 0.84(15) | 0.84 | 0.77(20) | 0.77(20) | 0.77 | 0.97(18) | 0.97(18) | 0.97 | 0.97(17) | 0.97(17) | 0.97 | 1.0(11) | 1.0(11) | 1.0 | 0.97(15) | 0.97(15) | 0.97 | 0.91(20) | 0.91(20) | 0.91 |
| MI | **0.99(17)** | **0.99(17)** | **0.99** | **0.98(2)** | **0.98(17)** | **0.98** | 0.8(13) | 0.8(13) | 0.8 | 0.75(3) | 0.75(3) | 0.75 | 0.94(18) | 0.94(18) | 0.94 | 0.97(11) | 0.97(11) | 0.97 | 1.0(12) | 1.0(12) | 1.0 | 0.97(17) | 0.97(16) | 0.97 | 0.91(19) | 0.91(19) | 0.91 |
| ls_l21 | 0.82(18) | 0.78(18) | 0.8 | 0.92(17) | 0.91(17) | 0.91 | 0.7(14) | 0.69(14) | 0.69 | 0.67(20) | 0.67(20) | 0.67 | 0.96(20) | 0.96(20) | 0.96 | 0.9(16) | 0.9(16) | 0.9 | 0.91(19) | 0.91(19) | 0.91 | 0.77(18) | 0.77(18) | 0.77 | 0.83(19) | 0.83(19) | 0.83 |
| ll_l21 | 0.91(19) | 0.9(19) | 0.9 | 0.87(14) | 0.86(14) | 0.86 | 0.76(20) | 0.76(20) | 0.76 | 0.73(19) | 0.73(19) | 0.73 | 0.96(16) | 0.96(16) | 0.96 | 0.91(18) | 0.9(18) | 0.9 | 0.97(17) | 0.97(17) | 0.97 | 0.85(20) | 0.85(20) | 0.85 | 0.78(20) | 0.78(20) | 0.78 |
| RFS | 0.87(14) | 0.85(14) | 0.86 | 0.96(19) | 0.96(19) | 0.96 | 0.68(12) | 0.69(12) | 0.68 | 0.69(20) | 0.67(20) | 0.68 | 0.95(20) | 0.95(20) | 0.95 | 0.93(19) | 0.91(19) | 0.92 | 0.94(20) | 0.93(20) | 0.93 | 0.85(19) | 0.85(19) | 0.85 | 0.79(20) | 0.79(20) | 0.79 |
| LASSO | 0.87(16) | 0.87(16) | 0.87 | 0.72(16) | 0.71(16) | 0.71 | 0.78(18) | 0.78(18) | 0.78 | **0.8(18)** | **0.78(18)** | **0.79** | 0.94(17) | 0.94(17) | 0.94 | 0.89(20) | 0.88(20) | 0.88 | 0.97(19) | 0.97(19) | 0.97 | 0.84(20) | 0.85(20) | 0.84 | 0.73(20) | 0.73(20) | 0.73 |
| EN | 0.87(16) | 0.87(16) | 0.87 | 0.72(16) | 0.71(16) | 0.71 | 0.78(18) | 0.78(18) | 0.78 | **0.8(18)** | **0.78(18)** | **0.79** | 0.94(17) | 0.94(17) | 0.94 | 0.89(20) | 0.88(20) | 0.88 | 0.97(19) | 0.97(19) | 0.97 | 0.84(20) | 0.85(20) | 0.84 | 0.73(20) | 0.73(20) | 0.73 |
| SMBA-CSFS | 0.83(16) | 0.83(16) | 0.83 | 0.86(20) | 0.86(20) | 0.86 | 0.67(20) | 0.68(20) | 0.67 | 0.8(20) | 0.77(20) | 0.78 | **0.98(15)** | **0.98(15)** | **0.98** | **0.99(19)** | **0.99(19)** | **0.99** | 1.0(20) | 1.0(20) | 1.0 | **0.99(20)** | **0.98(20)** | **0.98** | **0.97(20)** | **0.97(20)** | **0.97** |
| BSL | 1 | 1 | 1 | 1 | 1 | 1 | 0.74 | 0.74 | 0.74 | 0.92 | 0.92 | 0.92 | 0.93 | 0.93 | 0.93 | 0.8 | 0.8 | 0.8 | 1 | 1 | 1 | 0.98 | 0.98 | 0.98 | 1 | 1 | 1 |

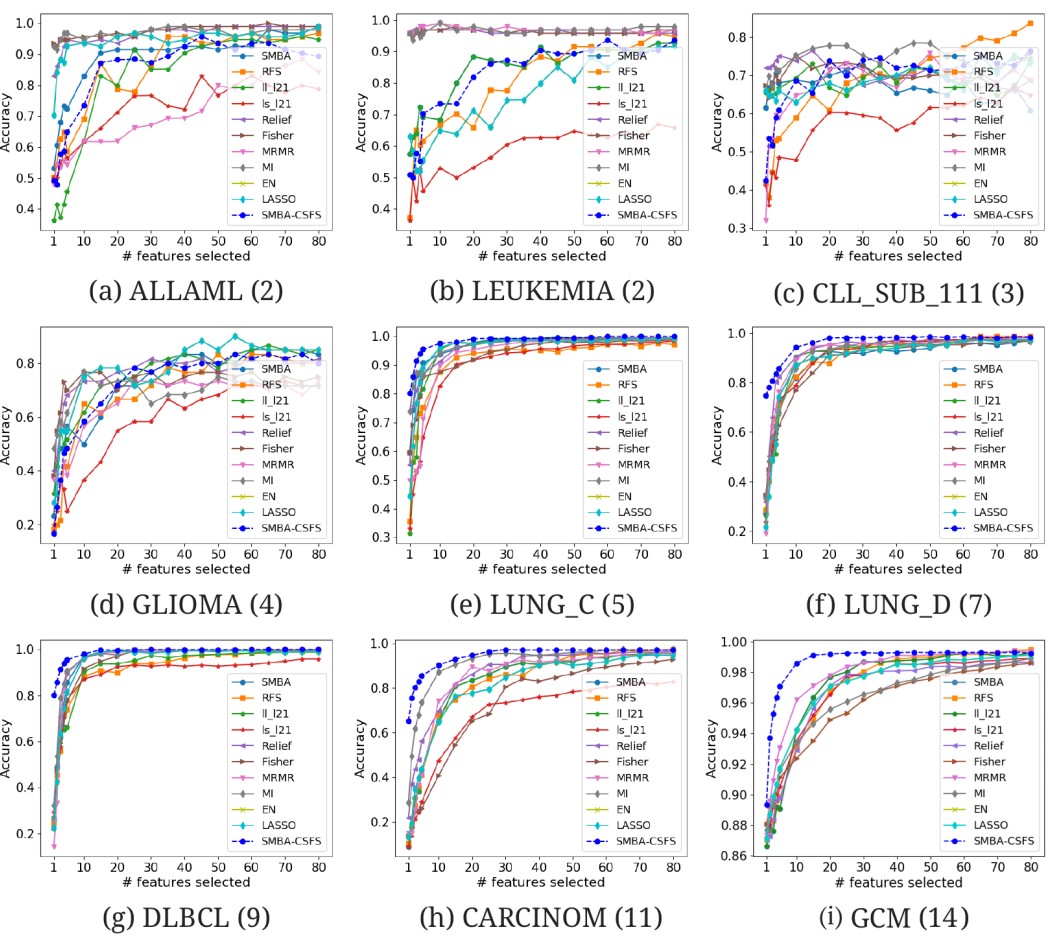

**Figure 2** **Comparison of several TFS accuracies against SMBA and SMBA-CSFS on nine data sets: (A) ALLAML(2), (B) LEUKEMIA(2), (C) CLL_SUB_111(3), (D) GLIOMA(4), (E) LUNG_C(5), (F) LUNG_D(7), (G) DLBCL(9), (H) CARCINOM(11), (I) GCM(14), when a varying number of features is selected.** SVM classifier with 5-fold CV was used.

sparse-modeling approach in selecting the features and the use of an ensemble classifier. Indeed, since the ensemble is based on a majority voting schema, SMBA-CSFS is able to guess, with higher probability, the belonging of samples coming from data sets with many classes. Just think that, whenever our method draws from a sample of a two-class data set, the probability of a right guess is proportional to a coin toss. Therefore if, on one hand, this leads to good performance when the data set consists of many classes, the probability of failure, on the other hand, increases in the case of data sets consisting of fewer classes. Anyhow, the local structure of data distribution which is crucial for feature selection, as stated in *He, Cai & Niyogi (2005)*, may be a logical reason why the SBMA schema performs better on certain data set rather than others. In addition, as shown in Fig. 2, it is worth observing that SMBA-CSFS seems to perform better w.r.t. TFS competitors on a fewer number of features. This would suggest that SMBA-CSFS is able to identify/retrieve the most representative features that maximize the classification accuracy. To assert the

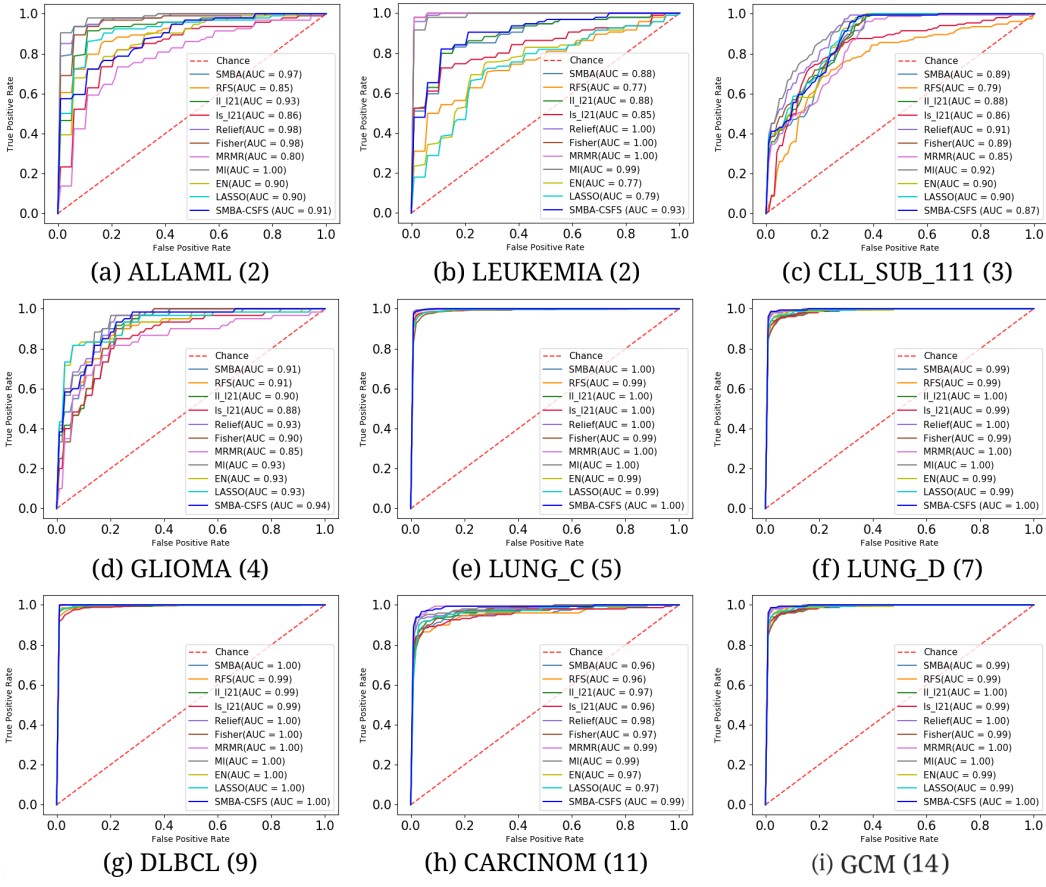

**Figure 3** **Average ROC curves and the corresponding AUC values on the first 20 features comparing the classification performance among SMBA-CSFS and TFS methods for nine data sets: (A) AL-LAML(2), (B) LEUKEMIA(2), (C) CLL_SUB_111(3), (D) GLIOMA(4), (E) LUNG_C(5), (F) LUNG_D(7), (G) DLBCL(9), (H) CARCINOM(11), (I) GCM(14).** SVM classifier with 5-fold CV was used.

previous results achieved, we computed the average ROC curves between SMBA-CSFS and the other TFS methods on a subset of 20 and 80 features, respectively. Looking at the AUC values in Fig. 3, it would suggest SMBA-CSFS as the best model to choose for identifying the most representative features in a classification task when dealing with data set with many classes. Concerning with the GF-CSFS competitors, as shown in Fig. 4, it would suggest that the *sparse modeling* process, underlying the proposed SMBA scheme for feature selection, is more suitable for retrieving the best features for the purpose of classification, often leading to get satisfactory results. Such statement is also proved by the good balance between precision and recall shown in Table 5 and the average ROC curves shown in Fig. 5, where SMBA-CSFS still holds a candle w.r.t. GF-CSF methods. The reader's attention is drawn to the Supplemental Material for all the experimental results and consideration arisen on the top 80 features.

To statistically validate the results and compare all the competing classifiers against the proposed SMBA-CSFS, on both 20 and 80 feature subsets, we ran *Non-Parametric multiple*

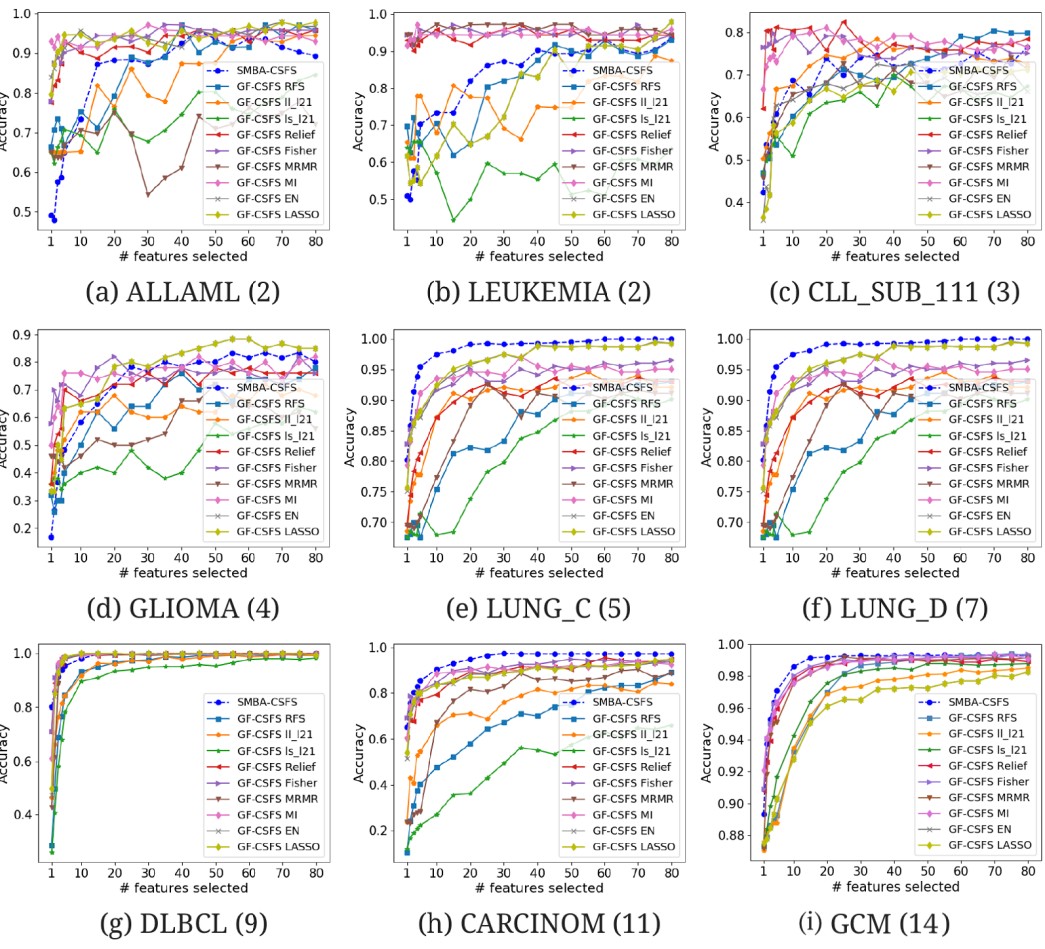

**Figure 4** **Comparison of several CSFS accuracies against SMBA-CSFS on nine data sets: (A) ALLAML(2), (B) LEUKEMIA(2), (C) CLL_SUB_111(3), (D) GLIOMA(4), (E) LUNG_C(5), (F) LUNG_D(7), (G) DLBCL(9), (H) CARCINOM(11), (I) GCM(14), when a varying number of features is selected.** SVM classifier with 5-fold CV was used.

comparison tests (all vs all) (*Demšar, 2006*; *Rodríguez-Fdez et al., 2015*) which sequentially performs a popular multi-class *Friedman nonparametric test* (*Friedman, 1937*) followed by a Nemenyi *Post-hoc multiple comparison* (*Dunn, 1961*). The ranking of the classifiers, when the top 20 and 80 features are selected, along with the corresponding p-values, are described in the Supplemental Material. Looking at the *Cumulative Rank* (CR) for each classifier, one can notice how SMBA-CSFS achieves optimal results, always finishing in the first three positions. However, it is worth emphasizing that our method ranks systematically on the top position when considering data sets consisting of five or more classes (named $CR_{\geq 5}$). These results prove again that SMBA-CSFS achieves good performance on data sets with many classes. Moreover, by using different classifiers we do not observe noteworthy differences in the results, meaning that the methodology is suitable for the classification of this kind of data, independently from the selected classifier. However, by looking at the *p*-values, corresponding to the single ranking method, one can better verify which algorithms have

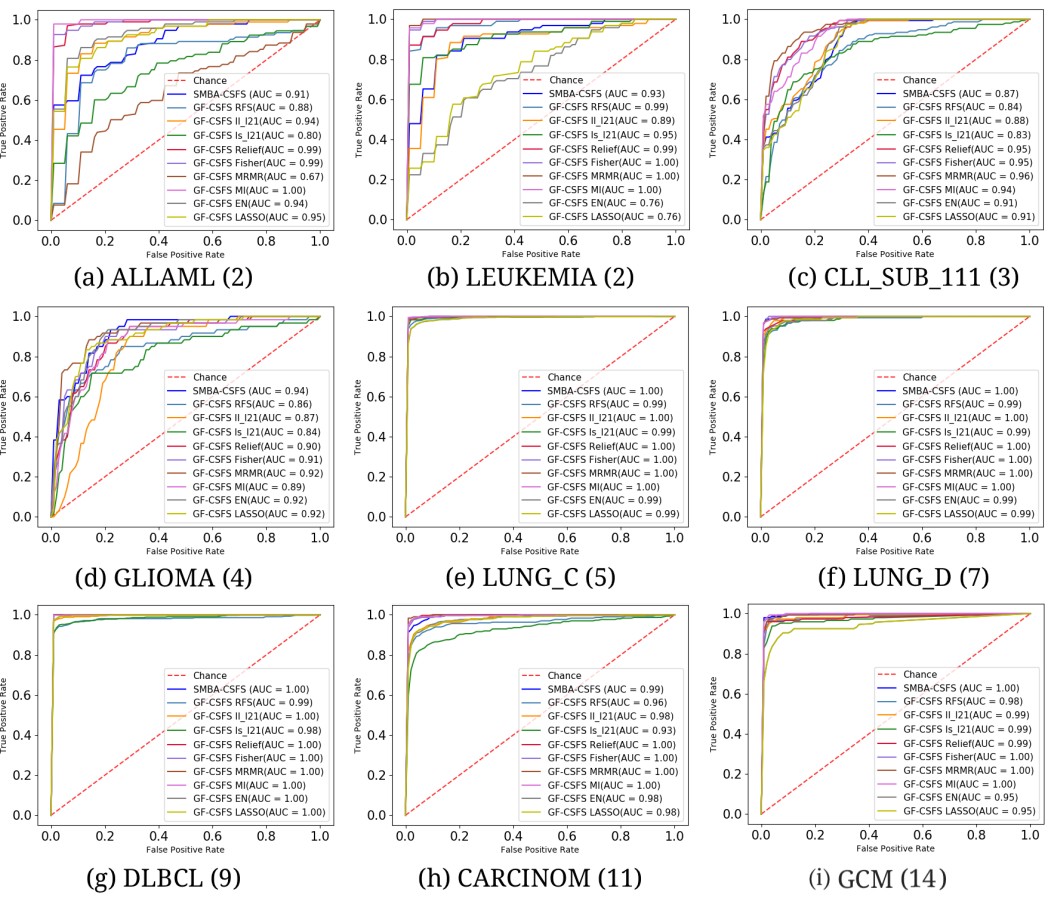

**Figure 5** Average ROC curves and the corresponding AUC values on the first 20 features comparing the classification performance among SMBA-CSFS and several CSFS methods for nine data sets: (A) ALLAML(2), (B) LEUKEMIA(2), (C) CLL_SUB_111(3), (D) GLIOMA(4), (E) LUNG_C(5), (F) LUNG_D(7), (G) DLBCL(9), (H) CARCINOM(11), (I) GCM(14). SVM classifier with 5-fold CV was used.

significantly different performance w.r.t. SMBA-CSFS. For detailed information regarding the results, see the Supplemental Material. Concerning the computational complexity, from several conducted experiments we observed that the proposed methodology may be slower than other techniques (e.g., FS and Relief whose running times are in term of few seconds) but comparable with SMBA. Its running time, depending on several parameters involved, especially in the size of the number of instances and classes of the data sets, may vary from a couple of hours to at most one day (see Table S9 for details on the computational time). Nevertheless, SMBA-CSFS achieves appreciable performance when working on large data sets and number of classes, and sometimes, in the biological field, the accuracy in finding key features that are responsible for some biological processes is preferred to the execution time. However, since most of the time consumed by the proposed approach is due to the solution of the optimization problem by using the ADMM method, and because the

methodology is based on an ensemble of classifiers, a parallel computing approach could be adopted to obtain a faster computational time (*Deng et al., 2017*).

## CONCLUSIONS

We proposed a Sparse-Modeling Based Approach for Feature Selection with emphasizing joint $\ell_{1,2}$-norm minimization and the Class-Specific Feature Selection. Experimental results, on nine different data sets, validate the unique aspects of SMBA-CSFS and demonstrate the promising performance achieved against the-state-of-art methods. One of the main characteristics of our framework is that, by jointly exploiting the idea of Sparse Modeling and Class-Specific Feature Selection, it is able to identify/retrieve the most representative features that maximize the classification accuracy in those cases where a given data set is made up of many classes. Based on our experimental results, we can conclude that, usually applying TFS allows achieving better results than using all the available features. However, in many cases, applying the proposed SMBA-CSFS method allows improving the performance of just TFS as well as GF-CSFS injected with several TFS methods. It has to be stressed, that SMBA-CSFS seems actually suitable for large data sets consisting of many classes, while on data sets with less than five classes other methods appear to be more effective. Although SMBA, SMBA-CSFS and TFS performance slightly differ on the whole, it is worth highlighting that SMBA-CSFS achieves its best performance when considering fewer features (i.e., from 1 to 20) on data sets with many classes, which is an important goal when certain biological tasks are taken into account. However, we do believe that these techniques might be effectively used in a systematic way after a microarray analysis. Indeed, a better gene selection step could avoid the waste of many resources in post-array wet analysis (e.g., Real Time-PCR) allowing researchers to focus their attention just on relevant features. Finally, we think this method demonstrated to be an interesting alternative among FS approaches on microarray data.

As future work, the focus will be moved towards the biologic interpretations of the SMBA framework behavior, by systematically studying the selected genes, especially taking into account the SMBA-CSFS approach which, as proved by the experimental results, is more effective in selecting genes of interest than the standard SMBA. Furthermore, we are planning to test our approach on EPIC data set (*Demetriou et al., 2013*), after a thorough analysis of pre-filtering, and a parallel implementation to substantially reduce its computational time.

## ACKNOWLEDGEMENTS

The research was entirely developed when Davide Nardone was a Master Degree student in Applied Computer Science at University of Naples Parthenope.

### Funding

This work was supported by Dipartimento di Scienze e Tecnologie Università degli Studi di Napoli "Parthenope" (Sostegno alla ricerca individuale per il triennio 2016–2018 project). The funders had no role in study design, data collection and analysis, decision to publish, or preparation of the manuscript.

### Grant Disclosures

The following grant information was disclosed by the authors:
Dipartimento di Scienze e Tecnologie Università degli Studi di Napoli "Parthenope" (Sostegno alla ricerca individuale per il triennio 2016–2018 project).

### Competing Interests

The authors declare there are no competing interests.

### Author Contributions

- Davide Nardone conceived and designed the experiments, performed the experiments, analyzed the data, prepared figures and/or tables, performed the computation work, authored or reviewed drafts of the paper, approved the final draft.
- Angelo Ciaramella and Antonino Staiano conceived and designed the experiments, prepared figures and/or tables, authored or reviewed drafts of the paper, approved the final draft.

### Data Availability

The data supporting the experiments in this article are available at Zenodo: Davide Nardone. (2019). Biological datasets for SMBA (Version 1.0.0). http://doi.org/10.5281/zenodo.2709491.

A Python software package is available through GitHub at https://github.com/DavideNardone/A-Sparse-Coding-Based-Approach-for-Class-Specific-Feature-Selection, containing all the source codes used to run the software.

### Supplemental Information

Supplemental information for this article can be found online at http://dx.doi.org/10.7717/peerj-cs.237#supplemental-information.

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
