# Peer review of "A Sparse-Modeling Based Approach for Class Specific Feature Selection"

_PeerJ Computer Science, doi:10.7717/peerj-cs.237_

## Round 0.1 · original submission · Major Revisions

I have received the review reports for your paper submitted to PeerJ Computer Science from the reviewers. According to the reports, I recommend major revision to your paper. Please refer to the reviewers’ opinions to improve your paper. Please also write a revision note such that the reviewers can easily check whether their comments are fully addressed. We look forward to receiving your revised manuscript soon.

Reviewer 1 ·

Basic reporting

The subsection: "A Sparse-Modeling Based Approach for Class-Specific Feature Selection" is actually too descriptive, while instead, a more algorithmic presentation would better suit to understand the proposed approach. I would suggest that the Authors still keep the current explanations, but that they also add the actual algorithm for SMBA-CSFS in pseudocode, naturally, with explanation of the symbols used in the algorithm. This should certainly be corrected.

Supplementary Table 2 caption: "The different colours represent the rank positions obtained: red first position, blue second position and green third position" should be removed from the caption.

Language issues and typos are present occasionally in the paper. It should be proof-read by the Authors.
For example:
45: "For these reasons, it is unsuitable using microarray data as they are" -> "For these reasons, it is unsuitable to use microarray data as they are"
82: "An active developing field of statistical learning is around the notion" -> "An active developing field of statistical learning is focused around the notion"
155: "4,026genes." -> "4,026 genes."
204: "Sparse-Modeling Based Approach is nothing else that our SMBA-CSFS model but that only take into" -> "Sparse-Modeling Based Approach is nothing else than our SMBA-CSFS model, but that only takes into"

Experimental design

99: "Once the class sample set of the training set has been split apart, it may be possible that each class-subset results unbalanced. Therefore, the SMOTE (Chawla et al., 2002) re-sampling method is applied to balance each class-subset."

The Authors do not describe how the splitting is performed. Since, if you split the dataset into subsets that belong to each class, then each class-subset contains only the samples of that class. There are no samples of the other classes. Why is then SMOTE needed, this makes no sense, or am I missing something?

118: "O will belong to the class that received more votes among the tied classes."

This is unclear. If the classes are tied, then they have the same number of votes. Then how can a class have more votes than others?

137: "The number of genes that are used in the multiclass classification task"

There are two classes, i.e. this is a binary classification problem, not a multiclass one.

Validity of the findings

227: "SMBA-CSFS is able to better discriminate among the classes of the LUNG C,LUNG D, CAR, DLBLC data sets in both cases, when top 20 and 80 features are considered."

Actually, for DLBCL, the claim can only be made for the first case, because in the second case, at least two other methods appear to be slightly better for 20 features. This is a minor issue.

Additional comments

The paper is mostly well-written, interesting and detailed in the performed analysis. The methods used are sound. I would recommend its acceptance after the corrections are made according to the provided comments.

·

Basic reporting

Although the paper is well written but organization is not good. English is OK. Sufficient references are included.

Experimental design

The motivation of the work is not mentioned clearly. The author should give a brief problem definition.

Although you have compared the method with existing methods such as mRMR, relief-F you can compare it with PSO-based, ACO-based feature selection methods also.

Validity of the findings

The experimental results look satisfactory but more experiments for statistical validation is required.

T-test is needed to compare the proposed method with other competing methods.
Test the method with some other high dimensional datasets.
The authors report accuracy only but other validation measures such as precision, recall and ROC curve should be given.

---

## Round 0.2 · Minor Revisions

Dear Professor:

I have received a re-review report for your paper submitted to PeerJ. According to the report, I recommend minor revision to your paper to give you the opportunity to edit the English language further.

With best regards
Tzung-Pei Hong
Associate editor of PeerJ

Reviewer 1 ·

Basic reporting

Only minor English language errors are still present, see below. Otherwise, the manuscript is well-written and structured.

Lines 14-15: "... and might speed the experimental validation up." -> "... and possible speed-up of the experimental validation"

In SMBA algorithm pseudocode:
- in line "Output", there should be "set of features", not "set of feature"
- there is a closing parenthesis missing in line 5

In "Algorithm 1":
- y, m, C and K should be in italic
- in line 10, there should be "subset of features selected", not "subset of feture selected"

Line 132: "O" should be in italic

Line 139: "the most m representative" -> "the m most representative"

Line 183: "190 samples in 14 classes, breast..." -> "190 samples in 14 classes: breast..."

Line 255: "f-measure" -> "F-measure"

Experimental design

No comment.

Validity of the findings

No comment.

Additional comments

The authors have answered satisfactory to all my queries.

---

## Round 0.3 · accepted · Accept

I am happy to recommend acceptance of the paper.